# Fish By-Product Use as Biostimulants: An Overview of the Current State of the Art, Including Relevant Legislation and Regulations within the EU and USA

**DOI:** 10.3390/molecules25051122

**Published:** 2020-03-03

**Authors:** Moses Madende, Maria Hayes

**Affiliations:** The Food BioSciences Department, Teagasc Food Research Centre, Ashtown, Dublin 15, Ireland; moses.madende@teagasc.ie

**Keywords:** biostimulants, fish protein hydrolysates, biostimulant regulation

## Abstract

Crop production systems have adopted cost-effective, sustainable and environmentally friendly agricultural practices to improve crop yields and the quality of food derived from plants. Approaches such as genetic selection and the creation of varieties displaying favorable traits such as disease and drought resistance have been used in the past and continue to be used. However, the use of biostimulants to promote plant growth has increasingly gained attention, and the market size for biostimulants is estimated to reach USD 4.14 billion by 2025. Plant biostimulants are products obtained from different inorganic or organic substances and microorganisms that can improve plant growth and productivity and abate the negative effects of abiotic stresses. They include materials such as protein hydrolysates, amino acids, humic substances, seaweed extracts and food or industrial waste-derived compounds. Fish processing waste products have potential applications as plant biostimulants. This review gives an overview of plant biostimulants with a focus on fish protein hydrolysates and legislation governing the use of plant biostimulants in agriculture.

## 1. Introduction

To reduce the number of toxic nitrates in the soil, the EU Council Directive of 12 December 1991 concerning the protection of waters against pollution caused by nitrates from agricultural sources (91/676/EEC) called for a significant reduction in the amount of nitrogen-containing fertilizers used in agriculture and horticulture. As a result, organic farming using natural preparations that improve the general health, vitality, and growth of plants and do not pose many environmental and ecological risks are now the preferred choice [1]. As an alternative to chemical fertilizers, bio- stimulants can be used to stimulate plant growth and increase yields [2]. Biostimulants are products able to act on the metabolic and enzymatic processes of plants improving productivity and crop quality. Biostimulants may also help plants to cope with stressful environmental conditions such as drought, abiotic stress and cold [3]. Plant biostimulants are defined as products that stimulate plant nutrition processes independently of the product’s nutrient content, with the sole aim of improving any of the plant or plant rhizosphere nutrient use efficiency, tolerance to abiotic stress, quality and availability of confined nutrients in soil or rhizosphere. Biostimulants may be any mixture of substances, organic in nature from natural or microbial sources that can improve crop conditions without causing adverse side-effects. Enzymes, proteins, amino acids and natural stimulants such as phenolics, fulvic acids and protein hydrolysates can be termed biostimulants. Fungi and microbes can be considered biostimulants and species, including *Trichoderma ree*sei and *Heteroconium chaetospira* and bacteria such as *Rhodococcus* sp. and Enterobacter sp. are used for this purpose.

Biostimulants are not fertilizers as they do not provide nutrients directly to plants, but they can facilitate the acquisition of nutrients by supporting metabolic processes in soil and plants. Biostimulants can be either plant extracts such as Rosemary or are often of animal origin. Animal sources of biostimulants are usually hydrolysates of food by-products such as casein [4], fish waste [5] or animal tissue [6]. They are ordinarily produced using either alkaline or chemical hydrolysis methods [2]. Chemical fertilizers and biostimulants can cause damage to the environment and the EU recommends that natural biostimulants will replace these in time. Biostimulants are also the preferred option of organic farmers and informed consumers [7,8]. Within the EU, regulation (EU) 2019/1009 of the European Parliament and of the Council of 5 June 2019 (http://data.europa.eu/eli/reg/2019/1009/oj) governs the suitability of by-products for use as either fertilizers or biostimulants. Plant biostimulants can be grouped on the basis of their origin, for example, humic substances (humic acids, fulvic acids, and humins), seaweed extracts, protein hydrolysates (plant and animal origin), beneficial microorganisms (bacteria and fungi), chitosans, silicon, and extracts from food waste or industrial waste streams among others [9].

Food waste streams are an important precursor for biostimulant development, and biostimulants have been developed from food waste streams, composts, manures, vermicompost, aquaculture, and fish processing waste streams and sewage treated products [3,10]. Animal processing by-products are ordinarily converted into protein hydrolysates using chemical or enzymatic hydrolysis for use as biostimulants. A protein hydrolysate was generated previously from chicken feathers and used as a biostimulant on maize [11]. Siapton^®^ is a product developed by controlled hydrolysis of organic substances of animal origin that is used as a foliar spray to prevent osmostress-related metabolic changes in maize [12]. Fish protein hydrolysates are also used as biostimulants and are ordinarily derived from fish skins and other by-products such as heads, muscle, viscera, bone, frames and roe [5]. Methods of identification of biostimulants generated from fish and animal by-products include amino acid analysis, eco-toxicological tests and sodium dodecyl sulphate polyacrylamide gel electrophoresis. Table 1 shows some of the commercially produced plant biostimulants from different sources that are currently available on the market for use in horticulture.

The active agents of protein hydrolysates are fat free amino acids including aspartic acid, hydroxyproline, threonine, serine, glutamic acid, proline, glycine, alanine, methionine, isoleucine, Leucine, tyrosine, melatonin, organic matter, short-chain peptides, and proteins [5]. Fish hydrolysates are proven to improve the utilization of nutrients by the plants and induce morphological changes in root architecture [16]. They may also have an anti-drought effect and may stimulate the growth and activity of beneficial microbes and improve antioxidant activity. The biological effects of these mechanisms of action are better root growth and development, increased root and leaf growth, induction of flowering and improved fruit setting, and reduced fruit drop [10]. The value of the European biostimulant market was USD 0.60 billion in 2018 and the US is the biggest consumer and producer of biostimulants. The global market for biostimulants is set to grow to USD 2.91 billion by 2021, with a compound annual growth rate of 10.4% from 2016–2021 [10].

## 2. Protein Hydrolysates

Diverse agricultural activities generate organic waste, which has the potential for further processing to produce biostimulants. The demand for biostimulants for use in modern agriculture has been driven in part by the need for fertilization with compounds of natural origin that are produced in an eco-friendly and sustainable manner [3]. In addition to being one of the fastest-growing animal food production systems worldwide, aquaculture is also becoming the main source of aquatic animal food for human consumption [17]. However, large quantities of fish by-products are also obtained from these processing streams. For example, over 60% of fish biomass, including head, skin, bones, tail, fins, viscera and whole fish rejects ends up as part of the fish processing waste stream [5]. Furthermore, fish processing operations produce large quantities of wastewater which contains varying degrees of organic contaminants and requires expensive processing procedures before being discarded. Only about 30% of the 91 million tons of fish harvested every year is transformed into fishmeal. Possibly more than 50% of the remaining fish tissue is processing waste and not used as food. The EU contributes approximately 5.2 million tons of discards per year [18]. As a result, this may cause pollution if improperly disposed of in the environment. Since most fish ‘waste’ consists of a significant amount of protein and fat [19], recycling by composting may constitute an important resource for biostimulant production. To facilitate the recovery of essential nutrients, fish processing by-products can be transformed into value-added products such as proteins, gelatin, oils, amino acids, minerals, hydrolysates and peptides through chemical or enzymatic hydrolytic means.

Fish skin is a very rich source of collagen and gelatin which can be hydrolyzed to gelatin hydrolysates and a variety of studies have been conducted to demonstrate the potential of fish skin from different species as a source of protein hydrolysates [5]. Wasswa et al. (2007) reported protein hydrolysate production from skin of grass carp fish using Alcalase [20]. Furthermore, Yin et al. (2010) reported the hydrolysis of catfish skin and subsequently described the rheological and functional properties of the generated protein hydrolysate [21]. In another study, Sampath Kumar et al. (2011) reported on the generation of protein hydrolysates from horse mackerel and croaker, by using pepsin, trypsin and α-chymotrypsin [22]. Animal-derived gelatin has been applied in agriculture as a plant biostimulant [23], as a result, fish skin as a waste product of fish processing with a high content of collagen can be hydrolyzed into gelatin and may be used for this purpose also. Fish processing also produces a large amount of fish heads as processing by-product waste. These can also be converted into protein hydrolysates using enzymatic hydrolysis. For example, Gbogouri et al. (2004) hydrolyzed salmon heads using Alcalase at optimum temperature, substrate: enzyme ratio and pH to produce various hydrolysates using a response surface hydrolysis approach [24]. Furthermore, Sathivel et al. (2003) showed significant hydrolysis of the head, whole fish, body and gonad of herring fish using Alcalase as the hydrolyzing enzyme with incubation for 75 minutes and described the functional properties of the produced protein hydrolysates [25]. Fish backbone for example has approximately 30% protein. As a result, several fish bone hydrolysis studies have been successfully carried out. A variety of studies have also demonstrated the hydrolysis of fish bone and frame into protein hydrolysates. Je et al. (2007) reported the protein hydrolysates from tuna backbone protein using different proteases such as Alcalase, α-chymotrypsin, Neutrase, papain, pepsin and trypsin [26]. In another study, Nazeer et al. (2011) reported hydrolysates from the backbones of Seela and ribbon fish using papain, trypsin and pepsin [27]. 

Similarly, fish frame—which is normally discarded during processing—can be hydrolyzed into protein hydrolysates. Liaset et al. (2000) used the industrial enzymes Neutrase, Alcalase and pepsin over a range of temperatures and times to hydrolyze Atlantic cod and Atlantic salmon fish frames without heads [28]. Hou et al. (2011b) successfully hydrolysed Alaska pollock frame by using Alcalase, Flavourzyme, acid protease, Protamex, alkaline protease, trypsin, MEAP, neutral protease, bromelain and papain commercial proteases [29]. Fish liver and eggs have also been reported as sources of protein hydrolysates. Chalamaiah et al. (2010) reported protein hydrolysates from ray-finned fish eggs using Alcalase and papain [30]. In another study, Ahn et al. (2010) reported protein hydrolysates from Tuna liver by protease hydrolysis using various proteases such as Alcalase, Neutrase and Protamex [31]. Several commercial products from fish protein hydrolysates mentioned above are available in the market. These include products such as custom collagen^®^ utilized by the pharmaceutical and cosmetic industry, hydrolyzed fish collagen type 1 capsules used in therapeutics, Levenorm^®^ antihypertensive capsules, Norland hydrolyzed fish collagen with pharmaceutical and food application, Protizen^®^ powder with anti-stress effects and Seagest™ capsules for dietary supplementation [32]. 

Enzymatic hydrolysis, using enzymes such as alcalase, papain, pepsin, trypsin, chymotrypsin, pancreatin, flavourzyme, pronase, neutrase, protamex, bromelain, cryotin F, protease N, protease A, orientase, thermolysin, and validase, is the most preferred method during the production of bioactive hydrolysates [5]. The production of hydrolysates via enzymatic hydrolysis occurs under controlled conditions of pH and temperatures and has been shown to have several advantages compared to chemical hydrolysis particularly with regards to the quality, bioactivity, and bioavailability of the end product [33,34]. Hydrolysis breaks down larger proteins into smaller soluble peptide chains containing 2–20 amino acids. Protein hydrolysates are classified according to whether they are of animal- or plant origin. Animal-origin protein hydrolysates include leather by-products, blood meal, fish by-products, chicken feathers and casein, whereas plant-origin protein hydrolysates include legume seeds, alfalfa hay and vegetable by-products [35].

Fish hydrolysates have shown excellent physicochemical and functional properties compared to their synthetic substitutes. Properties such as anti-oxidative activity [36], anti-hypertensive activity [37], antimicrobial activity [38] and anti-anemia activity [39] have been reported in the literature. Recently, researchers have developed a key interest in the anti-proliferative property of fish hydrolysates. Furthermore, fish protein hydrolysates have found possible application on a wide range of agriculture crops as biostimulants [40]. Protein hydrolysates as biostimulants are supplied as liquid extracts, soluble powders, and granular forms, and can be applied to the roots or as foliar sprays [35]. Globally, protein hydrolysates for agricultural use are produced by companies in Italy, Spain, the United States, China and India as well as Ireland, and were born from the leather and meat industries as an approach to valorize by-products.

### 2.1. Fish Hydrolysates as Plant Biostimulants

Fish by-products are enriched in proteins, fat, and amino acids following protease hydrolysis. Furthermore, fish by-products contain antioxidants, which are often suitable for food or feed applications depending on the method of storage and handling of the by-products [41]. The above-mentioned nutritional qualities make fish protein hydrolysates excellent candidates for use in organic agriculture as biostimulants like other animal-derived protein hydrolysates. Fish-derived protein hydrolysates as biostimulants were found previously to increase leaf numbers significantly, as well as the stem diameter, shoot and, root mass and succulence of a variety of plants. Additionally, the biostimulants enhanced leaf chlorophyll content, photosynthesis, and gas exchange and have the potential to be used for the sustainable production of lettuce [42]. Short-term (30-days after transplanting) effects of fish-derived protein hydrolysates, applied as a drench (3 mL·L^−1^ at 0, 14, and 24 d after transplanting), on soil properties and lettuce (Lactuca sativa L.) growth and physiology were evaluated in a growth chamber study. After harvesting, soil treated with protein hydrolysates had higher C/N ratio and content of K and Fe, lower pH values, electrical conductivity, water holding capacity and cation exchange capacity, and lower contents of NO_3_-N, P, Mg, SO_4_, Cu, Mn, B and Na, than soil of the control. Application of protein hydrolysates increased lettuce leaf numbers, stem diameter, shoot fresh and dry mass, and root dry mass [42]. It also increased leaf relative water content and succulence but did not affect specific leaf area. Protein hydrolysates increased leaf chlorophyll content, photosynthetic rate, stomatal conductance and transpiration, although they did not alter chlorophyll fluorescence. All results indicated that plant biostimulants (protein hydrolysates) from fish are effective tools for the sustainable production of lettuce [42]. In another study, Bhaskar et al. (2008) reported the enzymatic hydrolysis of visceral waste protein from Catla catla (Major south Asian Carp fish) using the Alcalase^®^ enzyme from Bacillus licheniformis [43]. The protein hydrolysate contained arginine, asparagine/aspartate, glutamine/glutamate, glycine, alanine, and proline/hydroxyproline. Kechaou et al. (2009) also reported a protein hydrolysate obtained from fish viscera (Sepia officinalis and Sardina pilchardus) using commercial enzymes from microorganisms, such as Alcalase^®^ and Flavourzyme^®^ (Novozymes/DK) [44]. According to the free amino acid composition, the results also indicated the potential use of the hydrolysate as a supplement for use in animal diets [44].

The growth of several plants is enhanced by the application of protein hydrolysates. Previously application of protein hydrolysates was shown to improve the growth of lettuce [37], winter wheat [45], tomato [46], corn [47] and peppers [48]. The application of ready-for-uptake amino acids in the form of protein hydrolysates allows plants to save energy on amino acid synthesis and increases the pace of their reconstruction, particularly during critical times of plant development such as after transplantation, during the flowering period and during times of climatic stress or plant diseases [49]. Amino acids can also act as chelators of metal ions; completely chelated mineral ions are neutralised, which can accelerate metal ions absorption and transport within the plant and thus enhance plant growth. Furthermore, increased growth of plants in the presence of biostimulants can be attributed to improved nutrient uptake, metabolism resulting from increases in soil microbial activity, root length, density and number of lateral roots, and increases in activities of enzymes involved in nutrient metabolism [35,50]. 

However, other studies have shown that protein hydrolysates may have negative or no positive effects on plant growth. For example, Lisiecka and co-workers showed that the application of an animal-derived biostimulant did not have a beneficial influence on the number of strawberry runners, their length, and their diameter. Furthermore, the biostimulant did not increase the number of strawberry daughter plants, their crown diameter as well as the number of leaves but instead decreased the weight of the strawberry plants significantly [51]. These contradictory data reveal that the beneficial effects of protein hydrolysates on plants are dependent on the sources (plant or animal origin) and preparation methods. Moreover, variations among studies could be attributed to the different application methods; plant species hydrolysates are applied to, and/or growth control conditions used.

Several protein hydrolysates of plant and animal origins have been made commercially available. C FISH is an example of a commercial protein hydrolysate that is derived from fish and is used as a plant biostimulant in Ireland (http://www.cfish.ie/plant-biostimulants/). Protein hydrolysates have shown a variable but significant increase in the yield and quality of agricultural and horticultural crops depending on the source [52]. Table 2 shows some examples of animal-derived protein hydrolysates whose potential as plant biostimulants has been studied. Corte et al. (2014) assessed the safety of hydrolysed proteins of animal-origin using yeast and plant models and concluded that they were no genotoxicity, ecotoxicity, or phytotoxicity observed, thus making them safe for use [53]. Fish protein hydrolysates contain many bioactive peptides and amino acids that have been shown to have human health and animal feed supplementation benefits (Table 3). In a related manner, these bioactive peptides may have positive effects on plants when applied as biostimulants. The peptide and amino acid composition of fish hydrolysates per 100 g protein is higher than that of casein, Alfalfa hay, chicken feathers and soya bean meal, but less than that of blood meal and bovine collagen [35] and these protein sources have been applied in horticulture as plant biostimulants. Considering the above-mentioned protein comparison and nutraceutical properties of fish protein hydrolysates, they potentially provide an excellent opportunity for widespread use in horticulture as plant biostimulants. Despite the aforementioned report, the EU has banned the application of animal-derived protein hydrolysates on the edible parts of organic crops including fish protein hydrolysates and fish meal, through the Commission Implementing Regulation (EU) no 354/2014 with regard to organic production, labeling and control (http://data.europa.eu/eli/reg_impl/2014/354/oj). However, animal-derived biostimulants can still be applied to no-edible parts of horticulture crops such as seeds during planting, roots and leaves to equally exert their biostimulant effect.

### 2.2. Individual Amino Acids

Individual amino acids from fish can stimulate plant growth. There is considerable evidence that the exogenous application of structural and non-structural amino acids such as glutamate, proline, histidine and taurine can provide protection to the plant from environmental stresses or play a role in metabolic signaling by regulating nitrogen acquisition by roots [76,77]. Specific effects of individual amino acid on plants are similar in many respects to those describe for protein hydrolysates. Amino acids have been shown to increase the uptake of nutrients by plants and to increase yields. For example, the product Amino16^®^, a protein hydrolysate containing 11.3% L-amino acids increased tomato yields compared to a control [78]. 

In other studies, by Walch-Liu et al. [79,80], the external application of L-glutamate on *Arabidopsis thaliana* plants showed that it can act as an exogenous signal to modulate root growth and branching [79,80]. Moreover, proline, betaine and their derivatives and precursors have also been shown to stimulate plant defenses to biotic and abiotic stress [81]. These amino acids act as osmo-protectants that stabilize proteins, enzymes and membranes against denaturing which is caused by high salt concentrations and non-physiological temperatures. Additionally, arginine has been shown to play an important role in nitrogen storage and transport in plants especially during abiotic and biotic stress conditions [82]. Amino acids can also decrease plant toxicity by heavy metals mostly by acting as metal chelators. There is substantial evidence that asparagine, glutamine and cysteine are important in Zn, Ni, Cu, As and Cd chelation [83,84].

## 3. Humic Substances

Humic substances are naturally occurring organic compounds that are produced from the degradation of plant material from terrestrial or marine sources and constitute a large reservoir of organic N and C. These substances are categorized into three groups based on their solubility at specific pH values. Humic acids are soluble at higher pH but insoluble at pH values less than 2. Fulvic acids are soluble at all pH values whereas humin is insoluble at any pH [85]. It has been shown that humic substances can enhance plant root growth and development by stimulating the uptake of nutrients, water and enhancing tolerance to environmental stress [86]. Humic substances have oxygen-, nitrogen- and Sulphur-containing functional groups which allow them to make stable complexes with metal micronutrients such as Fe, thereby facilitating the maintenance of micronutrients in their bioavailable forms [87]. 

In the same way, humic substances can also promote plant growth through the induction of carbon and nitrogen metabolism [88]. A variety of studies on the application of humic substances on several fruit crops, ornamental plants, and vegetables have been reported in the literature. To highlight a few, Ibrahim and Ramadan. (2015) [89] showed that the application of humic acids on common beans increased yield by 25–35%. In another study, Bettoni et al. (2016) showed that the application of humic substances on onion increased fresh root weight by 42–102% [90]. In contrast, Hartz and Bottoms. (2010) concluded that humic substances had no positive effects on nutrient uptake or productivity of vegetable crops [91]. 

## 4. Seaweed Extracts

Seaweeds form an integral part of coastal marine ecosystems and include a vast group of macroscopic, multicellular marine algae that are common inhabitants of the coastal regions of the world’s oceans. Macroalgae are broadly classified into brown, red, and green algae and are an important source of organic matter and fertilizer nutrients [92]. Several commercial seaweed extract products are available on the market for use in horticulture and agriculture and numerous studies have shown that seaweeds extracts can improve plant growth, productivity, yield, resistance to disease-causing microorganisms, abiotic stress tolerance and photosynthetic activity [93]. Literature highlights several studies in which seaweed extracts, and whole seaweed biomass or seaweed meal, were applied exogenically to plants to evaluate their biostimulant effects. For example, Carvalho et al. (2013) showed that seaweeds extract increased germination in bean plants [94], whereas Abdel-Mawgoud et al. (2010) showed that seaweeds extracts increase the yield of watermelon [95]. Seaweed extracts application has also been shown to enhance the shelf life of spinach in addition to improving flavonoid synthesis and nutritional quality [96,97]. The growth effects of seaweed extracts are believed to be due to complexes of carbohydrates, minerals, and trace elements as well as growth regulatory compounds such as phytohormones contained in seaweeds [98].

## 5. Microorganisms

Microorganisms such as bacteria, yeast, filamentous fungi, and micro-algae have been shown to have biostimulant activity. Arbuscular mycorrhizal fungi have many symbiosis-associated benefits to plants such as efficient nutrient use, water balance, biotic, and abiotic stress protection of plants, and as a result, it has been used as a plant biostimulant [99]. Arbuscular mycorrhizal fungi-based products have been applied to plants to promote nutrition efficiency by absorbing and translocating mineral nutrients beyond the depletion zones of plant rhizosphere and induce changes in secondary metabolism. Additionally, these products also increase tolerance to stress, crop yield, product quality, and influence the phytohormone balance of host plants, thereby influencing plant development [16,100]. *Trichoderma*-based products have received attention as biostimulants due to their capacity to control phytopathogenic fungi. Phytostimulation by *Trichoderma* involves the release into the rhizosphere of auxins, small peptides, volatiles and other active metabolites, which promote root branching and nutrient uptake, and ultimately enhancing growth, development and adaptation to abiotic stress [101]. Plant growth-promoting rhizobacteria have been used in horticulture to promote plant growth as a biostimulant. Its mode of action includes improvement in the availability of nutrients, the production of volatile organic compounds, hormone release and hormonal changes within plants, and the enhancement of tolerance to abiotic stresses [102]. An example includes the exogenous application of *Rhizobium rubi* to broccoli, which increased yield, plant weight, head diameter, chlorophyll content, macronutrient, and micronutrient uptake [103].

## 6. Production, Composition and Quality Control of Commercial Biostimulants

A wide variety of technologies are used in the production and preparation of biostimulants both alone and in combinations. These include chemical and enzymatic hydrolysis, extraction, cultivation, fermentation, high pressure cell rupture and processing among many others [10]. However, many manufacturers tend to not disclose biostimulant production technologies used to protect their commercial secret [104]. Depending on the manufacturing conditions and treatments used in biostimulant production, some commercial biostimulants may contain compounds that are not initially present in the raw material. Thus, commercial biostimulants often marketed as equivalent products, may differ considerably in composition and efficiency [105]. Biostimulants contain both primary metabolites such as amino acids, sugars, nucleotides, and lipids and secondary metabolites such as aromatic amino acids, phenolic compounds, terpenoids/isoprenoids, alkaloids, and glucosinolates [106]. The presence of secondary metabolites is highly dependent on the raw material used in the production of the biostimulant. In most cases, biostimulants are composed of multiple components such as plant hormones or hormone-like substances, amino acids, betaines, peptides, proteins, sugars (carbohydrates, oligo-, and polysaccharides), aminopolysaccharides, lipids, vitamins, nucleotides or nucleosides, humic substances, beneficial elements, phenolic compounds, furostanol glycosides, and sterols [10].

A variety of other compounds that are possibly present in biostimulant products have not yet been characterized and still many others are not known whether they retain their activity following processing during production. In order to maintain and ensure consistence in biostimulant product quality, many methods are used to qualitatively and quantitively analyze the compounds present in commercial biostimulants products [10]. These methods include amino acid analysis, bioassays, ecotoxicological tests, Fourier transform infrared spectroscopy (FTIR), gas chromatography coupled with mass spectrometry (GC/MS), sodium dodecyl sulphate polyacrylamide gel electrophoresis (SDS PAGE), 13C NMR, 1H NMR, atmospheric pressure chemical ionization-mass spectrometry (APCI-MS), cross-polarization magic angle spinning (CP/MAS), (CPMAS)-13C-NMR, diffuse-reflectance infrared Fourier transform spectroscopy (DRIFT), electronic microscopy, elemental analysis, high performance liquid chromatography tandem mass spectrometry (HPLC/MS/MS), pyrolysis-gas chromatography-mass spectrometry, UV–vis and ELISA [107]. However, maintaining the consistence in the quality of biostimulant products remains challenging.

Fish protein hydrolysate for use as plant biostimulants or for other uses is produced either as the liquid form which contains up to 90% moisture or as a dry form. The dry form is usually preferred due to its stability during long-term storage and easier transport, however, removing large amounts of water during the production of dried fish protein hydrolysates can be difficult and very costly [108]. The manufacture of fish protein hydrolysates often begins with the collection of raw materials, mainly fish or fish processing left-over products. This is followed by solubilization with water typically in the ration of 1:1 and a hydrolysis step in metallic tanks called hydrolysators. Hydrolysis could either be chemical (acid or alkali) or enzymatic (commercial or fish indigenous proteases) hydrolysis. The main aim of hydrolysis is to extract fish proteins into low molecular weight peptides and individual amino acids [109]. The temperature and time used in hydrolyzation is dependent on the type of hydrolysis, the nature of the raw material, the chemicals used and the nature of the final product and its intended use. When the intended degree of hydrolyzation is reached, the process of hydrolyzation is terminated chemically or by adjusting the temperature [110].

Hydrolyzation is followed by a dehydration process which encompasses many other pre-drying treatments. Firstly, the hydrolyzed mixture is purified by the removal of insoluble and fat fractions via centrifugation and plate and frame filtration. Fish oil fraction is reduced to a final concentration of 0.5% to avoid undesirable fat oxidation processes in the final product [111]. Further fractionation or concentration of the recovered hydrolyzed fish proteins is achieved by micro-, ultra- and nano-filtration. Membrane filtration is particularly useful tool for obtaining bioactive peptides from fish protein hydrolysates [112]. Filtration is followed by a concentration step in dryers, the protein hydrolysate solution can be concentrated up to 50% solids [113]. Finally, steps of fish protein hydrolysate include drying, packaging and storage/transport. The type of drying depends on the availability of equipment and energy savings. Usually, spray- and freeze-drying techniques are used, although in some cases roller drum drying can be a method of choice for drying [108]. Fish protein hydrolysates are usually packaged and stored in a dry place at 4 °C before distribution.

## 7. Plant Biostimulants’ Mode of Action

The mode action of biostimulants ranges from activation of nitrogen metabolism or phosphorus release from soils to the stimulation of root growth and enhanced plant establishment as detailed in the above sections [10]. However, in many cases the mode of action of biostimulants is not known, causing disputes over the legitimacy of commercialized biostimulant products [114]. Biostimulants enhance nutrient uptake and assimilation by plants. This is often attributed to the capacity of biostimulants to increase the activity of soil both microbiologically and enzymatically and to alter the root structure and change the solubility and transportability of micronutrients [35,47]. Plant biostimulants may increase the amount of nutrients available to plants by increasing soil cation exchange via nitrogen provision and enhancing solubility of soil nutrients [115]. Some biostimulants make complexes with insoluble elements such as Fe and make them available for plants. Humic compounds lower the pH of soil and root surface by increasing the activity of plasma membrane H^+^-ATPase which releases H^+^ into the soil, lower soil pH enhances the availability and uptake of nutrients [116].

As discussed in the previous sections, protein hydrolysates are an important group of plant biostimulants. Fish protein hydrolysates are a mixture of oligopeptides, polypeptides and amino acids that can be applied as foliar sprays or dosed into the soil near the plant’s roots [35]. In addition to enhancing soil properties like respiration, protein hydrolysates act as growth stimulants for soil microorganisms that can utilize them as easy source of carbon and nitrogen. Protein hydrolysates can also complex and chelate soil micro- and macronutrients such as Fe so that these become more accessible to plants [13]. In a study by Trevisan et al (2017), a novel biostimulant APR (collagen derived protein hydrolysate), showed a significant enhancement of the dry weight of both roots and root/shoot ratio and mRNA-Sequencing analysis revealed transcriptional changes in root of maize seedlings [117]. In a follow up study, Trevisan et al. (2019) elucidated the mechanism of action involved in the biostimulant effect of APR^®^ on maize seedlings under abiotic stress. The most significant effect of APR^®^ on growth re-establishment was observed for roots, which are the main target for hypoxia, salt and nutrient deprivation stresses [118]. Additionally, the study showed a marked regulation of the transcription of genes encoding members of the high affinity nitrate transport system (*HATS*, *NRT2* and *NAR* genes), which was particularly relevant in condition of abiotic stresses. The study also showed that APR^®^ might preventively prepare plants to oxidative stresses by the regulation of ROS signaling genes, in this study *SOD1A* gene was clearly regulated in the presence of stress and APR. In another study by Wilson et al. (2015), gelatin capsules were applied in soil near cucumber seeds and caused the increase in fresh and dry weight biomass, leaf area and nitrogen content of 2-week old plants. These improvements were credited to an upregulation of both amino acids and N transporter genes and the xenobiotic detoxification system [119].

## 8. Regulation of Plant Biostimulants in Europe and USA

The regulation of biostimulants is generally very complicated and, until recently, remained in search of an official identity. This is mainly due to the lack of a formal and precise definition and agreement on what encompasses a biostimulant among the different regulatory bodies [16]. There are currently two ways in which biostimulants are introduced in the market in Europe, these include either following national regulations on fertilizers or the European pesticides law, which combines both supranational and national provisions for introducing plant protection products on the market. The EC regulation No 1107/2009 on plant protection products is applicable to all categories of biostimulants in Europe. As a result, synthetic and natural substances (including botanicals and basic substances as mentioned before), and microorganisms, are all covered by this regulation due to their non-nutrient stimulation of plant growth (http://data.europa.eu/eli/reg/2009/1107/oj ). All plant growth regulators and herbicide safeners, which are substances that interact with the physiology of the plant, even though they do not protect the plant against pests or diseases, have been registered under the EC regulation No 1107/2009 until recently. Due to the growing concerns pertaining to the process and the cost of registering a plant protection product on the European market, biostimulants can be alternatively regulated under the EC fertilizers regulation (regulation (EC) No 2003/2003) (http://data.europa.eu/eli/reg/2003/2003/oj). 

However, in 2019, the European Commission established regulations that state which types of bio-products can be used in organic farming under the fertilizing products regulation (EU) 2019/1009. Fertilizing products include fertilizers as well as other categories of products such as biostimulants and growing media (http://data.europa.eu/eli/reg/2019/1009/oj). The Fertilizing Products Regulation, which fully applies from dates subsequent to the 16th of July 2022, will provide common rules on safety, quality, and labeling requirements for all fertilizing products to be traded freely across the EU. It will open the market for products, which are not currently covered by harmonization rules, such as organic and organo-mineral fertilizers, soil improvers, inhibitors, plant biostimulants, growing media or blends. Among some of the materials consisting of EU fertilizing products are by-products within the meaning of the Waste Framework Directive (Directive 2008/98/EC). This Directive lays down measures to protect the environment and human health by preventing or reducing the generation of waste, the adverse impacts of the generation and management of waste and by reducing overall impacts of resource use and improving the efficiency of such use, which are crucial for the transition to a circular economy and for guaranteeing the Union’s long-term competitiveness (https://eur-lex.europa.eu/legal-content/EN/TXT/?uri=CELEX:02008L0098-20180705 ).

The European Union (EU) Fertilizing Products Regulation (EU, regulation (EU) 2019/1009) proposes a claim-based definition of biostimulants, stipulating that “plant biostimulant” means a product stimulating plant nutrition processes independently of the product’s nutrient content, with the aim of improving one or more of the following characteristics of the plant: (1) nutrient use efficiency, (2) tolerance to abiotic stress, (3) crop quality traits, or (4) availability of confined nutrients in the soil and rhizosphere [120]. The future regulation also specifies that a plant biostimulant “shall have the effects that are claimed on the label for the plants specified thereon.” This creates an onus for manufacturers to demonstrate to regulators and customers that product claims are justified. Consequently, the justification of the agronomic claim of a given plant biostimulant will be an important element to allow it to be placed on the EU market. The European Biostimulant Industry Council (EBIC) proposes some general guiding principles to follow when justifying plant biostimulant claims. These principles are expected to be incorporated into harmonized European standards that are being developed by the European Committee for Standardization (CEN) to support the implementation of the regulation.

In order to support a biostimulant claim, published literature evidence can be provided to demonstrate product’s characteristics such as mode of action if it is of acceptable quality. However, synergistic effects of compounds that may occur in a proposed product with biostimulant effects means that literature evidence alone may not be enough [121]. As a result, experimental data can be used to complement literature evidence with field trials providing essential information about the biostimulant where possible. Furthermore, the net agricultural benefit after considering both the positive and negative effects of the biostimulant should be large enough to justify its use. Lastly, general guidelines for conducting trials or assays for biostimulants are available and must cover components that include the aim of the trial series, statistical analysis and trial design, trial conditions, design and lay-out of trials, control data, application of treatments, and mode of assessment [120].

In the US, the biostimulant market has grown exponentially over the past decade and, according to a new report by Fortune business insights, the global biostimulant size market size is projected to reach USD 5.69 Billion by the end of 2026 [122]. Like Europe, the regulation of biostimulants in the US is still not clearly defined and thus threatens to slow down its expected growth. On the 20th of December 2018, the 2018 Farm Bill was passed into law as the Agricultural Improvement Act and had recommendations regarding plant biostimulants [123]. According to the Bill, a biostimulant is described as “a substance or micro-organism that, when applied to seeds, plants or the rhizosphere stimulates natural processes to enhance of benefit nutrient uptake, nutrient efficiency, tolerance to abiotic stress, or crop quality and yield.” The regulation of biostimulants is dependent on the active components of the product and the claims made for it. Thus, a plant biostimulant could either be considered a plant regulator under the Federal Insecticide, Fungicide and Rodenticide Act (FIFRA), leading to its regulation by the Environmental Protection Agency (EPA). Alternatively, the biostimulant can be regulated by state departments of agriculture that regulate fertilizers, plant and soil amendments, and other products not regulated by FIFRA. Following the 2018 Farm Bill, in March 2019, the United States EPA released draft guidance on plant biostimulants entitled: Guidance for Plant Regulator Label Claims, Including Plant Biostimulants (https://www.federalregister.gov/documents/2019/03/27/2019-05879/pesticides-draft-guidance-for-pesticide-registrants-on-plant-regulator-label-claims-including-plant) [124]. The draft describes a plant biostimulant as “a naturally-occurring substance or microbe that is used either by itself or in combination with other naturally-occurring substances or microbes to stimulate natural processes in plants or the soil to improve nutrient and or water use efficiency by plants, help plants tolerate abiotic stress, or improve the physical, chemical, and or biological characteristics of the soil as a medium for plant growth.” The draft also provides clarity on claims that are considered non-plant regulator claims and those that are plant regulator claims. Plant regulators can cause the acceleration, retardation, or modification of a plant, or produce growth unlike non-plant regulators, which merely improve conditions to aid in plant growth and nutrition. 

## 9. Conclusions

With the rising demand to meet the nutritional requirements of a growing population and widespread consumer awareness of environmentally-friendly agricultural practices as well as strict regulations on the use of chemical fertilizers, there is an urgent need to find alternative methods of sustainable horticultural production through technical and technological innovations. The use of plant biostimulants from fish by-products can improve nutrient uptake, nutritional efficiency, plant yields and the quality of products and is a promising alternative to conventional chemical fertilizer use. This has led to an increase in the demand for plant biostimulants and an exponential market growth, which is set to continue with the same growth trajectory over the next few years. 

The industrial processing of fish produces large amounts of by-products that can be transformed into value-added products such as plant biostimulants. The application of smaller quantities of fish protein hydrolysates in horticultural practices has resulted in increased crop yields and fruit and vegetable quality compared to chemical fertilizers in several studies [5,18,42]. Fish processing waste products are currently being transformed into protein hydrolysate liquids or solids. The processing involves several steps such as chemicals or enzymatic hydrolysis, filtration, drying and storage and many of the processing parameters of these steps are governed by the intended use of the final product. To enable consistent product quality, analytical test techniques on products such as SDS PAGE and NMR are often used. The regulation of biostimulants for use in horticulture has also been a topic of intense discussion. The EC and USDA have now provided a framework for the regulation and use of biostimulants in Europe and the US respectively. In line with the (EU) 2019/1009 regulation and the onus that it placed on manufacturers to justify a biostimulant claim, EBIC has proposed some general guiding principles to follow when justifying plant biostimulant claims [119].

The use of biostimulants in agriculture continues to increase in popularity, and thus more research is necessary to explore all the compounds contained in a variety of processing waste, such as in fish processing with biostimulant activity. Moreover, data from research will also provide further insights into the mode of action of biostimulants, which may open more avenues for specific biostimulant formulations as well as other possible agricultural applications. Recently, Futureco Bioscience launched a novel biostimulant Radisan^®^, that is rich in microelements, amino acids, peptides and phytohormone activators (https://www.futurecobioscience.com/en/products/biostimulants/#radisan). This is further testament to a growing interest in the use of plant biostimulants in horticulture; thus, the elucidation of their mechanism of action becomes critical. Research concerning wastes from dairy processing recently stressed that the timing, mode and frequency of application of biostimulants needs to be optimized for maximal effects on plant health. It is also clear that the benefits to plants may vary among years depending on weather and other factors [125].

## Figures and Tables

**Table 1 molecules-25-01122-t001:** Commercially available plant biostimulants, their composition and application strategies [13,14,15].

Biostimulant	Origin	Active Compounds	Application Methods	Plant	Main Activity
C Fish	White fish/mixed fish composition autolysates and hydrolysates	Peptides, amino acids	Foliar, irrigation, pre-planting	Vegetables, fruits	Increase plant’s resistance to insect pressure, disease and heat or drought stress
Radifarm	Commercial formulation	Amino acids, peptides, saponins, betaines, polysaccharides, vitamins, microelements	Irrigation, soil drench, foliar application	Fruits and vegetables	Promotes the formation of an extensive root system by speeding up the elongation of lateral and adventitious roots
Megafol	Commercial formulation	Amino acids, betaines, proteins, vitamins, auxin, gibberellin, cytokine	Irrigation, soil drench, foliar application	Fruits and vegetables	Promotes balanced vegetative development and productivity, and plant resistance to stress (frost, root asphyxia, weeding, hail)
Biozyme	Ascophyllum nodosum	Algae extract, plant hormones, chelated micronutrients	Irrigation, foliar, pre-planting, soil drench	Fruits, vegetables, legumes,	Increase nutrient uptake and activity of chlorophyll and photosynthesis
Algreen	Seaweed	Seaweed extract, plant hormones, vitamins, free amino acids, alginic acid			Promotes growth and yield parameters, enhance vitamin C and dry matter content
BioRoot	Plant derived protein hydrolysates	Plant and mineral-derived organic acids and humates, alfalfa and soybean meal, brewer’s yeast, K-sulfate, rock phosphate, sea kelp	Irrigation, foliar, soil drench	Fruits and vegetables	Increase rooting ability and chlorophyll and protein contents
Kelpak	Ecklonia maxima,	Seaweed extract	Drip irrigation, Soil drench, Seed treatment, foliar	Fruits and vegetables	Stimulate plant’s natural hormones, root initiation and germination
Biplantol Universal	Commercial formulation	Macro-and microelements, germanium, uronic acids, medicinal herbs, worm humus	Foliar, soil drench	Fruits, vegetables, flowers	Resistance to fungal diseases and insect pests
Grow-plex SP	Humic acids	Humic acids	Irrigation, foliar	Fruits, vegetables	Stimulate soil bacteria, root and shoot growth, iron and zinc uptake
Tablet	Microorganism	Rhizophagus intraradices and Trychoderma atroviride spores	Soil drench	vegetables	Stimulate root system architecture (higher total root length and surface), improve chlorophyll synthesis and increase proline accumulation
Ergonfill	Animal derived protein hydrolysates	Animal protein hydrolysates, cysteine, folic acid, keratin derivatives	Foliar	Fruits and vegetables	Promotes indolacetic acid and chlorophyll synthesis, improves translocation and chelation of macro and trace elements
Benefit	Commercial formulation	Amino acids, nucleotides, free enzymatic proteins, vitamins	Irrigation, foliar, soil drench	Fruits and vegetables	Stimulates cell division and increase in the number of cells per fruit

**Table 2 molecules-25-01122-t002:** Examples of protein hydrolysates from different fish and animal sources and their application as plant biostimulants.

Source	Plant	Growth Media	Method of Application	Bioactive Compounds	Biostimulant Activity	Reference
Fish-derived protein hydrolysate	Lettuce	Field soil	Exogenous (during watering)	Peptides, amino acids	Increased leaf number and root biomass Enhanced chlorophyll content, photosynthetic rate	[42]
Commercial amino acids preparation	Lettuce	Field soil	Foliar	Glycine and glutamine	Increased yield, leaf chlorophyll and vitamin C content	[54]
Meat flour protein hydrolysate	Maize	Hoagland solution	Seedlings immersed in solution	Small peptides and amino acids	Stimulation of root and leaf biomass Induced nitrate conversion to organic nitrogen Stimulate efficient nutrient utilization by plants	[47]
Commercial preparation of chicken feather hydrolysate	Wheat	Field soil	Foliar	Short peptides and amino acids	Increased yield and nutrient content of grains	[45]
Chicken feather hydrolysates	Maize	Field soil	Foliar	Peptides and amino acids	Increased micro- and macronutrient concentration of leaves Increased yield and grain protein content	[55]
Commercial amino acid preparation	Coriander	Hoagland nutrient solution	Dissolved into growth media	Glycine	Increased growth of roots and shoots Increased micronutrient content of leaves	[56]
Fe-amino acid chelates preparation	Tomato	Nutrient solution	Dissolved into nutrient solution	Arginine, glycine and histidine	Increased uptake of Fe and improved root and shoot growth	[57]
Commercial animal-derived calcium protein hydrolysate	Rojo Brillante	Field soil	Irrigation	Peptides, amino acids and metal elements	Lower chloride uptake and reduction in leaf necrosis	[58]
Commercial animal-derived amino acids product	Tomato	Nutrient solution	Foliar and root application	Amino acids	No effects on Iron nutrition Caused severe plant depression	[59] [60]
Commercial preparation of amino acids and peptides	Passion fruit	Commercial growing medium	Foliar	Amino acids and peptides	Promotes the photosynthetic process in plants Improved transplanting successes	[61]
Animal derived gelatin	Cucumber, pepper, broccoli, tomato, arugula, and field corn	Field soil	Exogenous (adjacent to seeds)	Amino acids and peptides	Increased shoot dry weight Increased root N assimilation	[23]

**Table 3 molecules-25-01122-t003:** Examples of fish protein hydrolysates used in animal feed supplementation and human health.

Protein Hydrolysate Source	Hydrolysis Method	Bioactive Compounds	Bioactivity	Reference
Pollock	Alcalase and flavourzyme	Short peptides	Growth by growth hormone stimulation	[62]
Pollock	Chemical (formic acid) and Enzymatic	Short peptides	Induce immune-modulatory effects enhancing survival	[63]
Commercial fish protein hydrolysate	Enzymatic	Glutamic acid, other amino acids and peptides	Contain opioid-like compounds that may have anti-stress effects	[64]
Barbel	Alcalase	Short peptides	Could be used as antimicrobials or antibiotic adjuvants	[65]
Half-Fin anchovy	Enzymatic	Bioactive peptides	Antibacterial activity	[66]
Pacific hake	Flavourzyme	Amino acids and peptides	Cryoprotectant	[67]
Catshark	Enzyme	Peptides	Emulsifying	[68]
Shark Capelin	Alcalase	Short peptides	Foaming	[69]
Salmon	Enzymes (Alcalase, Flavourzyme, Corolase)	Peptides and amino acids	Water binding	[70]
Sardine Tuna	Enzymes Alcalase, neutrase, papain, pepsin	Peptides and amino acids	Antioxidative	[71] [72]
Tuna	Alcalase, neutrase, Protamex	Peptides	Antihypertensive ACE-inhibitory	[73]
Yellow fin tuna	Protamex	Lower molecular peptides	Antimicrobial	[74]
Slender lizard fish	Papain	Peptides	Antianemia	[39]
Cod and Saithe fish meal	Protamex	Bioactive peptides	ACE inhibitory	[75]

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
