# Peer review of "Fish By-Product Use as Biostimulants: An Overview of the Current State of the Art, Including Relevant Legislation and Regulations within the EU and USA"

_molecules, 2020, doi:10.3390/molecules25051122_

Round 1
Reviewer 1 Report
Based on the papers reporting on biostimulants, there is a consensus to write it without hyphen. I suggest to keep consistency in the literature and use a single nomenclature. Europe legislation has adopted a precise definition in May 2019 and it would be helpful to keep the meaning of biostimulant closely adhered to the regulation and the business (line 42). The relevance of biostimulants finds its origin in the application and use in agriculture and not so much in research, simply because biostimulants were historically first developed and used in agriculture. The authors chose to classify biostimulant in 5 major classes, associated with the type or orgin of extracts used (line 60). The European legislator has actually classified according to the function of biostimulants. In my opinion this is a much more clear classification as new products (or even the well known silicon that does not fit the classification given) can readily be classified without creating new groups. Indeed, a few lines down, products are listed that do not match to any of the five major groups listed.
This review focusses on the potential use of fish by-products as a source for the production of biostimulants. This is a valid concept as indeed fish waste contains peptides, lipids, and other metabolites that may stimulate plant growth and a few examples have reported. However, the potential use for biostimulants competes with other attempts to valorise this biomass. The statement that fish waste is discarded as is, does not correspond to what generally is done (line104, and compare with what is written line 170-197 and in Table 3 where several applications are described). The waste is being processed to create many different products, not the least to produce fish meal as feed for fish culturing. The current use of fish waste can certainly improve, one of which is the development as biostimulant.
The authors refer to the various reports on the hydrolysis of fish waste (line 135-144). This supports the potential of fish waste to generate peptides and amino acids. However, it is not clear to the reader how these products are currently used or how they are commercialized.
It is likely that in comparison to the nutrional value of fish hydrolysate, biostimulants are economically not equivalent, most likely preventing the further exploitation. The key problem however, is, as stated in line 236 and onward, that animal derived protein hydrolysates can not be used for crop production. That simply limits the value of the review to the non-EU region.
The subsequent sections review biostimulants that are not related to fish by products. It was a surprice to read about humic acids and seaweed extracts, microorganims, etc in a review that focusses on fish byproducts.
In my opinion the idea to summarize and discuss sources for biostimulants is of interest, included a focus on fish products. I suggest that the authors include or animal protein as a source to discuss. But then including other sources does not make much sense. It would be of interest to learn how non-EU countries view the concept that biostimulants are generated from animal origin.
Author Response
Reviewer 1
Based on the papers reporting on biostimulants, there is a consensus to write it without hyphen. I suggest to keep consistency in the literature and use a single nomenclature.
Response: All hyphens on the word biostimulant have been removed
Europe legislation has adopted a precise definition in May 2019 and it would be helpful to keep the meaning of biostimulant closely adhered to the regulation and the business (line 42). The relevance of biostimulants finds its origin in the application and use in agriculture and not so much in research, simply because biostimulants were historically first developed and used in agriculture.
Response: This section has been re-written and the latest biostimulant definition has been taken into account
The authors chose to classify biostimulant in 5 major classes, associated with the type or orgin of extracts used (line 60). The European legislator has actually classified according to the function of biostimulants. In my opinion this is a much more clear classification as new products (or even the well known silicon that does not fit the classification given) can readily be classified without creating new groups. Indeed, a few lines down, products are listed that do not match to any of the five major groups listed.
Response: The classification criteria has been modified to include other biostimulants that do not fit in the previously mentioned major groups
This review focusses on the potential use of fish by-products as a source for the production of biostimulants. This is a valid concept as indeed fish waste contains peptides, lipids, and other metabolites that may stimulate plant growth and a few examples have reported. However, the potential use for biostimulants competes with other attempts to valorise this biomass. The statement that fish waste is discarded as is, does not correspond to what generally is done (line104, and compare with what is written line 170-197 and in Table 3 where several applications are described). The waste is being processed to create many different products, not the least to produce fish meal as feed for fish culturing. The current use of fish waste can certainly improve, one of which is the development as biostimulant.
Response: The authors have clarified the concern above
The authors refer to the various reports on the hydrolysis of fish waste (line 135-144). This supports the potential of fish waste to generate peptides and amino acids. However, it is not clear to the reader how these products are currently used or how they are commercialized.
Response: The authors have made an attempt to include data on the commercialization of the mentioned fish waste hydrolysis products
It is likely that in comparison to the nutrional value of fish hydrolysate, biostimulants are economically not equivalent, most likely preventing the further exploitation. The key problem however, is, as stated in line 236 and onward, that animal derived protein hydrolysates can not be used for crop production. That simply limits the value of the review to the non-EU region.
Response: This point has been expanded to give further insight and clarification on the issues that have been raised
The subsequent sections review biostimulants that are not related to fish by products. It was a surprice to read about humic acids and seaweed extracts, microorganims, etc in a review that focusses on fish byproducts.
Response: Although the review focused largely on fish biostimulants, the authors found it necessary to minimally discuss biostimulants from other sources (where extensive work has been done) in order to give the reader a overall perspective on plant biostimulants.
In my opinion the idea to summarize and discuss sources for biostimulants is of interest, included a focus on fish products. I suggest that the authors include or animal protein as a source to discuss. But then including other sources does not make much sense. It would be of interest to learn how non-EU countries view the concept that biostimulants are generated from animal origin

Reviewer 2 Report
The subject is of interest and in the scope of journal. The paper is well structured and includes original data that maybe of interest for the readers of Molecules.
Comments:
- comments were marked in the text of manuscript

Author Response
Reviewer 2
Table 1 need to complete with main activity (beneficial effects) of bio-stimulants in an additional column with references
Response. The main activity or beneficial effects of all the commercial plant biostimulants in table 1 have been added and references also provided
Reviewer 3 Report
The manuscript entitled "Fish by-product use as Bio-stimulants: An overview of the current state-of-the-art including relevant legislation and regulations within the EU and USA" presents an extensive review of the use of bio-stimulants with a particular focus on fish-derived products.
Overall the manuscript is well written, easy to follow and covers the most important aspects of the field.
I recommend this article for publication with the only concern on the length of the publication, which should be evaluated at the editorial level.
I found one typo on line 500, which should be addressed before publication: "The regulation With the rising demand".
Author Response
Reviewer 3
The manuscript entitled "Fish by-product use as Bio-stimulants: An overview of the current state-of-the-art including relevant legislation and regulations within the EU and USA" presents an extensive review of the use of bio-stimulants with a particular focus on fish-derived products.
Overall the manuscript is well written, easy to follow and covers the most important aspects of the field.
I recommend this article for publication with the only concern on the length of the publication, which should be evaluated at the editorial level.
I found one typo on line 500, which should be addressed before publication: "The regulation With the rising demand".
.
Response: Typo corrected